# Diarrhoeagenic *Escherichia coli* and *Salmonella* spp. Contamination of Food and Water Consumed by Children with Diarrhoea in Maputo, Mozambique

**DOI:** 10.3390/ijerph21091122

**Published:** 2024-08-26

**Authors:** Sara Faife, Custódia Macuamule, Josphat Gichure, Tine Hald, Elna Buys

**Affiliations:** 1Department of Consumer and Food Sciences, Faculty of Natural and Agricultural Sciences, University of Pretoria, Private Bag X20, Pretoria 0028, South Africa; slfaife@gmail.com (S.F.); jngichure@gmail.com (J.G.); 2Department of Hygiene and Public Health, Veterinary Faculty, Eduardo Mondlane University, Maputo P.O. Box 257, Mozambique; cushymacuamule@gmail.com; 3National Food Institute, Technical University of Denmark, Private Bag 201, 2800 Kongens Lyngby, Denmark; tiha@food.dtu.dk

**Keywords:** food, water, diarrhoeagenic *Escherichia coli*, *Salmonella* spp., Mozambique

## Abstract

In Mozambique, about 500,000 cases of diarrhoea were caused by foodborne pathogens in 2018. A review of the epidemiology of diarrhoea in children under five showed a high disease burden. This study aimed to identify Diarrhoeagenic *Escherichia coli* (DEC) and *Salmonella* spp. contamination of food and water in urban and rural areas of Maputo consumed by children under five with diarrhoea. One hundred and eighty-six children with diarrhoea were selected from Primeiro de Maio and Marracuene Health Care Centres from the Kamaxakeni and Marracuene districts, respectively. Food (n = 167) and water (n = 100) samples were collected in children’s households for diarrhoeagenic bacterial identification. Interviews were conducted using a semi-structured questionnaire to collect data about demographics and foods consumed a week before the children’s diarrhoea episodes. The prevalence of both DEC and *Salmonella* spp. was 9.8% in food and 5.4% in water samples. DEC was most prevalent in cereals (urban = 2.8%; rural = 2.4%) and water samples (urban = 1.4%; rural = 3.3%). *Salmonella* spp. was mainly detected in cereals (urban = 0.7%; rural = 0.8%). Diarrhoeagenic pathogens were associated with the type of food frequently consumed by children under five years with diarrhoea (infant formula, fruit puree, ready-to-eat meals, and bottled water), while the association with demographics was absent. We found that the infant foods consumed by children with diarrhoea are associated with DEC and *Salmonella* spp., and the prevalence of these contaminants is higher in the rural (8.9%) than in the urban area (6.3%), showing the need for caregiver education on food handling practices.

## 1. Introduction

Foodborne illnesses constitute a public health concern, leading to 420,000 deaths yearly worldwide, where low and middle-income countries have the highest burden, accounting for 53% of all foodborne illnesses and 70% of related deaths [1]. Children under five years are at higher risk, and almost 40% of foodborne diseases and 30% of associated deaths occur in this group age [2].

The high foodborne illness rates in low and middle-income countries can be attributed to several factors. These include the lack of clean water for washing utensils and food, unsafe water supply systems, poor sanitation, inadequate hygiene, and the use of human sewage or animal waste for irrigation [3]. Contaminants in food may be due to food preparation and storage, hygiene practices, children feeding themselves, and exposure to infected animals [4,5]. In Mozambique, about 500,000 cases of diarrhoea were caused by foodborne pathogens in 2018 [6]. A review of the epidemiology of diarrhoea in children under five years showed a high disease burden and an associated case fatality rate of 10% [7]. From 2015 to 2019, 9041 cases of diarrhoea were reported in children under five years old in Maputo, of which 25.2% were in rural areas and 74.8% were in urban areas [8].

The majority of the current literature available for Mozambique identified diarrheagenic pathogens in humans. These included Rotavirus, Adenovirus, *Shigella* spp., *Escherichia coli*, *Vibrio cholera*, and *Cryptosporidium* spp. [7,9]. Limited existing data showed an association between caregiver hygiene practices during food preparation and storage and infant food contamination [4]. This study aimed to ascertain the prevalence of diarrhoeagenic pathogens, such as *E. coli* and *Salmonella* spp., in food and water consumed by children under five years of age with diarrhoea.

## 2. Materials and Methods

### 2.1. Target Population

Children under five were selected from the Primeiro de Maio Health Care Centre in the Kamaxakeni district (urban area) and the Marracuene Health Care Centre in the Marracuene district (rural area) in Maputo (Mozambique). These health care centres offer primary health care services.

A total of 186 children were selected based on convenience sampling methods where they were included because they appeared in the health care centre and if they fulfilled the following inclusion criteria: (i) children were under five years old with diarrhoea, (ii) their caregiver consented to complete the questionnaire, and (iii) consented to provide food and water samples from their respective households for laboratory analyses.

### 2.2. Interviews with Caregivers

We conducted face-to-face interviews based on a semi-structured questionnaire to collect demographic data and information on the possible source/vehicle of infection as perceived by the caregivers. We also gathered data on the food consumed in the household a week before the children experienced diarrhoea episodes and the water supply. The questionnaire was designed to obtain information about the food consumed by children under five, especially those younger than two years old.

### 2.3. Food and Water Samples Collected from Households

A total of 267 samples, comprising 167 food and 100 tap water specimens, were collected from selected households. About 100 g of food that was available during sampling collection and 1L of water were collected with sterile containers, kept chilled and transported to the laboratory at Eduardo Mondlane University, Mozambique.

The breakdown of food samples included cereal (urban = 57; rural = 26), combined foods (urban = 30; rural = 24), pasteurised cow’s milk and milk products (urban = 8; rural = 3), cooked vegetables (urban = 5; rural = 2), and other miscellaneous foods (urban = 2) (Table 1). It is important to note that “combined food” typically entailed a blend of pap or rice with curry, or a meat or vegetable stew. This type of meal is prevalent in Mozambican communities as a complementary food for children transitioning to solid foods. Regarding the water samples, 42 were sourced from urban locations, while the remaining 58 were collected from rural settings.

### 2.4. Bacteriological Analysis of Food and Water Samples

#### 2.4.1. Detection of *Salmonella* spp., *Shigella* spp., and *E. coli*

Table 2 summarises *Salmonella* spp., *Shigella* spp., and *E. coli* methodology used. *Salmonella* spp. and *Shigella* spp. detection was conducted using the standard operating procedure created based on the Food and Drug Administration-Bacteriological Analytical Manual (FDA BAM) and ISO 6579-1:2017, while *E. coli* identification was conducted following the FDA-BAM and National Authority official methods [10,11,12,13]. From these analyses, 104 food and 37 water samples were found to have *Salmonella* spp./*Shigella* spp. isolates, while 49 food and 38 water samples harboured *E. coli* strains. A single colony of each isolate was further analysed using biochemical tests, confirming the identification of the strains obtained in the culture method.

#### 2.4.2. MALDI-TOF MS Confirmation of Bacterial Isolates

For confirmation, 228 isolates were submitted to the matrix-assisted laser desorption/ionisation time-of-flight mass spectrometry (MALDI-TOF MS) [14]. Single colonies from overnight-grown bacteria isolated from food and water samples were analysed on MALDI Biotyper^®^ sirius RUO (Bruker Daltonics GmbH & Co. KG, 2021; Bremen, Germany) using Bruker Daltonics MALDI-TOF biotyper flex control 3.4 software. A representative spectrum was obtained by collecting 40 shots on six different positions of the sample (240 laser shots in total) in about 15 s per sample, and the results were displayed in real time. The results were obtained by comparing the Bruker Daltonics MALDI-TOF biotyper compass RTC 4.1.100 database strains.

#### 2.4.3. DNA Extraction and Diarrhoeagenic *Escherichia coli* PCR

DNA was extracted from confirmed 16 h E. coli cultures using the Quick-DNA Fungal/Bacterial Miniprep kit (Zymo Research, Tustin, CA, USA).

Polymerase Chain Reaction (PCR) amplification of Diarrhoeagenic *Escherichia coli* (DEC) genes was performed using SSI Diagnostic protocol (Hillerød, Denmark), which detects Enterohemorrhagic *Escherichia coli* (EHEC), Enteroinvasive *Escherichia coli* (EIEC), Enteropathogenic *Escherichia coli* (EPEC), and Enterotoxigenic *Escherichia coli* (ETEC) pathotypes.

The PCR was carried out in a T100TM Thermal cycler (Bio-Rad, Hercules, CA, USA) in a total volume of 20 μL, containing 10 μL PCR ReadyMix, 6 μL Primer Mix, and 4 µL of extracted DNA. The amplification cycle included denaturation for 2 min at 95 °C, which was performed in one cycle, followed by 35 cycles of denaturation at 94 °C for 50 s, annealing for 40 s at 62 °C, extension at 72 °C for 50 s, and one cycle of the final annealing at 72 °C for 3 min.

The 18 µL of each completed PCR reaction were visualised in 2% agarose gel for 40 min at 70 V, stained with SYBR green, and detected by Gel doc TM EZ system (Bio-Rad, Hercules, CA, USA).

### 2.5. Data Analysis

Preliminary screening of the demographic characteristics and the caregiver’s perception of the cause of diarrhoea in the children was conducted using univariate analyses on Epi info (version 7.2.5.0) [15]. The univariate analysis screened the variables for inclusion in the binomial logistic regression model using X^2^ with the threshold set at *p* ≤ 0.2 with an odd ratio (OR > 1), based on the Fischer test and considering the one-tailed hypothesis. The binomial regression model was performed in the R environment for Windows (version 4.3.1) using the package MASS (Version 7.3-60) [16]. For the binomial regression analysis, the presence or absence of DEC/Salmonella in food samples was considered an outcome variable, while the explanatory variables included child age, caregiver age, child gender, caregiver gender, marital status, source of thinking, sampling site, baby bottle, bottled water, fruit puree, infant formula, milk breastfeeding, ready-to-eat meals, goat, cassava leaves, water source, water treatment, monthly income, and toilet system. The X^2^ test was conducted to detect differences between the urban and rural areas.

## 3. Results

### 3.1. The Sociodemographic Characteristics of the Caregivers and Children under Five Years Old

The study included 186 children, most of whom were from the urban area (54.8%) and mostly male (58.8%). In the rural area, most children were female (51.2%). The majority of the children (53.2%) were between 7 and 24 months old, and their caregivers were mainly female (94.1%), with over half of them being between 18 and 25 years old. The majority of the caregivers were single but living with a partner (76.8%) (Table 3). The X^2^ test indicated a significant difference (*p* < 0.05) in child age in the urban and rural areas.

In terms of monthly income, most caregivers in the urban area (33.3%) and the rural area (28.6%) were unaware of the primary breadwinner’s income. Among those who did provide information, the majority reported it to be between $71.00 and $140.00 (8.8% urban; 9.5% rural) (Table 3).

Concerning water supply, 99.0% of the participants used potable municipal water in the urban area, while in the rural area, they used municipal water and community wells. For water treatment, boiling was the most used method (10.8% urban; 8.3% rural). Flush or pour toilets with septic tanks, including squat toilets, were the most common toilet systems: 56.9% in urban areas and 32.1% in rural areas. The X^2^ test indicated a significant difference (*p* < 0.05) in the water source and toilet system used in both sampling areas (Table 3).

### 3.2. The Caregiver’s Perception of the Cause of Diarrhoea in Children under Five Years Old

Table 4 shows the cause of diarrhoea as perceived by the caregivers, where the majority of respondents in both urban, 52.0% (53/102), and rural, 25% (23/84), areas could not identify a probable cause for episodes of diarrhoea in the children included in the study. However, some caregivers attributed the cause of diarrhoea to factors such as vegetables, cereals, cookies/cakes/popcorn, weaning/infant formula, teething, and fruits/juice. Interestingly, only a small percentage of respondents in the urban area (2.9%) and the rural area (2.4%) considered water a probable cause of diarrhoea. The X^2^ indicated that there was a significant difference (*p* < 0.05) in caregivers’ perception of the cause of diarrhoea between the urban and rural areas.

### 3.3. One-Week Food Consumption Recall for Children under Five with Diarrhoea

Table 5 details the food consumed by children with diarrhoea the week before their visit to the health care centre. Based on the one-week food recall, the children had consumed animal protein, vegetables, and fruits (only rural). Chickens were the most commonly consumed source of animal protein, accounting for 36.3% of cases in the urban area and 90.5% in the rural area. Among dairy products, milk was reported as the most commonly consumed in the urban area, accounting for 12.7% of cases, while in the rural area, yoghurt was the most commonly consumed, accounting for 29.8% of cases. Cheese was only reported to be consumed by the children included in the study in the urban area, 2.9% of cases. Most children consumed green cabbage a week before the in 37.3% of the episodes of diarrhoea in the urban area and 19.0% in the rural area. Fruits were only consumed in the rural area, in 7.1% of cases.

Regarding food consumption by children under two years old, biscuits/rusks/cookies were the most commonly consumed item in the urban area, accounting for 33.3% of consumption. In the rural area, fermented cereals were the most consumed, representing 33.3% of consumption. Children under two years old were also fed liquid foods using feeding bottles, with 21.6% in urban areas and 29.8% in rural areas (Table 5).

The *p*-value of the X^2^ test indicated a significant difference (*p* < 0.05) in the consumption of certain foods before the occurrence of diarrhoeal episodes in children in both sampling areas. These foods include animal proteins (such as beef, pork, goat, chicken, eggs, fish, and yoghurt) and vegetables like cacana (*Momordica balsamina*), which is used as a food in south Mozambique and as medicine in other parts of the country (Table 5).

### 3.4. Prevalence of Foodborne Pathogens in Food and Water Samples

Table 6 shows the prevalence of foodborne pathogens in samples collected from the children’s households. In both rural and urban areas, contamination with Diarrhoeagenic *E. coli* (DEC) and *Salmonella* spp. was found in the food and water collected from the children’s homes. The overall prevalence of bacterial contamination was 15.2%, with a higher prevalence in the rural area (8.9%), and water and cereal samples contributed 12.1%. The most prevalent pathogen was DEC, primarily found in cereal and water samples, with a prevalence of 9.9% in both areas. *Salmonella* spp. was only identified in the water samples (0.7%) collected in the urban area and cereal samples (1.5%) from both areas. DEC was the only pathogen detected in combined food (2.4%) and yoghurt (0.7%) from rural and urban areas, respectively.

### 3.5. Prevalence of Diarrhoeagenic E. coli in Food and Water Samples

Table 7 represents the prevalence of DEC pathotypes among food and water samples collected in households and the sampling sites. The overall frequency of the DEC pathotypes is 13.0% and ETEC was the most prevalent, contributing 11.5% in both rural and urban areas. The other pathotypes consisted of EIEC detected in cereal food from rural areas and EPEC identified in water samples from urban areas.

### 3.6. Associated Factors

Table 8 presents the results of a binomial logistic regression analysis that aimed to assess the relationship between the prevalence of DEC isolated from the food and water samples and the demographics and food consumed by children participating in the study. However, this analysis was not conducted for *Salmonella* spp. because only three samples were found to be contaminated with this pathogen, which falls below the threshold for conducting binomial logistic regression.

The results of the analysis demonstrated a significant association between DEC contamination and the consumption of fruit puree [(B = 1.50); 95% CI (1.31–14.70), (*p* < 0.05)] and infant formula [(B = 1.34); 95% CI (0.91–13.90), (*p* < 0.05)] in children under two years of age. Additionally, the binomial logistic regression analysis conducted on the urban area data revealed associations between the consumption of certain foods by children under two years, such as associations of infant formula [(B = 2.24); 95% CI (0.04–83.40), (*p* < 0.05)] and ready-to-eat meals [(B = 2.03); 95% (0.84–60.7), (*p* < 0.05)] with DEC contamination. Conversely, in the rural area, the consumption of fruit puree [(B = 2.17); 95% CI (1.65–56.2), (*p* < 0.05)] and bottled water [(B = 3.04); 95% CI (2.79–>100), (*p* < 0.05)] by children under five years of age was associated with DEC contamination. Notably, no association was found between demographic characteristics and DEC contamination in this study.

## 4. Discussion

This study assessed the prevalence of diarrhoeagenic bacteria in food and water consumed by children with diarrhoea in urban and rural areas of Maputo and factors associated with children’s food contamination.

Most of the selected children were below two years old in urban and rural areas, showing that children of this age group were the most affected by diarrhoea [17].

The source of water in urban and rural households was different. In the urban area, only tap water was used, while in rural households, tap water and community wells were used as sources of water. This difference may be related to the coverage of municipal water distribution, which is high in the urban area [18]. The type of toilet facility used in the urban and rural areas is also different; more than half of households in the urban area use flush or pour toilets with septic tanks, including squat toilets. This may be related to lower socioeconomic status, as this type of facility is more costly than others.

Although caregivers from urban and rural areas indicated several factors as responsible for the cause of diarrhoea in children under five, most of them did not know the cause. Similar to a study carried out in Ethiopia [19], in this study, the percentage of respondents without knowledge of the cause of diarrhoea was higher in the urban area than in the rural area, contradicting the expected association with education level in this area. Chicken consumption in Maputo was lower than reported in other areas of Africa, which may be a protective factor, as poultry may carry pathogenic agents, including *E. coli*, *Listeria monocytogenes*, Campylobacter, Salmonella, and *Staphylococcus aureus* that can cause diseases, especially in children with weak immune systems [20,21,22].

Yoghurt consumption was higher in the rural area compared to a study conducted in Nigeria [23]. The consumption of yoghurt benefits for gastrointestinal conditions such as diarrhoeal disease [24]. The consumption of lettuce and banana was different in urban and rural areas; this fresh produce may constitute a vehicle for diarrhoeagenic pathogens when good hygiene is not appropriately followed [25].

DEC was the most frequent pathogen isolated in water collected from households in urban and rural areas (5.4%), which was lower than the prevalence reported in another study from Mozambique [26]. The presence of DEC in drinking water may be due to faecal contamination. *Salmonella* spp. was only detected in water samples collected in the urban area of Maputo, while in South Africa, this pathogen was identified in rural areas [27]. The presence of *Salmonella* spp. in water samples may be related to the dissemination of these bacteria through the faecal–oral route. Cereal-based foods were the most consumed by children in urban and rural areas, and DEC and *Salmonella* spp. were contaminants of this food group, similar to findings reported in South Africa. The contamination by DEC might have occurred during food preparation and the preservation of leftovers. *Salmonella* spp. contamination may be related to sanitation and poor operational practices during food processing [28,29]. The prevalence of diarrhoeagenic contaminants was higher in the rural area than in the urban area, which may be due to the ability of people in urban areas to purchase and access better quality and safe foods [30]. Additionally, the urban area (Kamaxakeni) has benefited from several educational sessions related to food handling practices, as this has been the focus of many studies that include health promotion activities to prevent illness. Contamination of combined food collected in households was observed, similar to a study in Bangladesh that reported contamination of complementary food with *E. coli* [31]. The presence of DEC in tested food samples indicates faecal contamination and poor hygiene conditions during food preparation and conservation, which can cause gastroenteritis in children [32].

The present study evaluated the association between diverse types of food and water with DEC contamination. Infant formula and fruit puree were associated with DEC contamination. Infant formula contamination by *E. coli* may occur during preparation, storage, and feeding, making this infant food unacceptable for consumption, so following hygiene practices is essential [33].

Unpasteurised fruit products can contribute to foodborne illness as they can act as vehicles for foodborne pathogens. For example, berry juices and purees have been shown to support pathogens like *E. coli* O157:H7 [25]. In the urban area, factors associated with the presence of contaminants were infant formula and ready-to-eat meals, which is concerning as improper practices during their preparation can introduce diarrhoeagenic pathogens to children.

In the rural area, food contamination was associated with fruit puree and bottled water. The use of bottled water was associated with DEC contamination, which may be related to water counterfeiting. Other studies have also found DEC in bottled water, showing that water that people believe to be safe for consumption can be a risk factor for foodborne illness [34].

The present study did not address aspects related to how the caretaker prepares the food and how this is conserved, especially the leftover food, which are crucial points where contamination can occur; that is why a study should be carried out to ascertain when and where the contamination takes place in Maputo households.

## 5. Conclusions

Diarrhoeagenic *E. coli* and *Salmonella* spp. are present in food and water consumed by children under five years with diarrhoea in Kamaxakeni, an urban area, and Marracuene, a rural area of Maputo. The prevalence of these diarrhoeagenic pathogens is higher in rural areas.

Significant disparities are noted between the two study areas in caregivers’ perceptions of the causes of diarrhoea, the types of water sources and toilet systems utilized, and the dietary habits of children in the week preceding diarrheal episodes.

These findings underscore the necessity for educating caregivers on proper food handling practices and enhancing water sources, given the evidence of diarrhoeagenic pathogens.

## Figures and Tables

**Table 1 ijerph-21-01122-t001:** Food and water samples collected from households of children with diarrhoea in urban and rural areas of Maputo.

Collected Samples	Sampling Site	
	Urban Area No (%)	Rural Area No (%)
Cereal	57 (21.3)	36 (13.5)
Combined food	30 (11.2)	24 (9.0)
Milk and milk product	8 (3.0)	3 (1.1)
Vegetable (cooked)	5 (1.9)	2 (0.7)
Other food (fish and fruit puree)	2 (0.7)	0
Water	42 (15.7)	58 (21.7)

**Table 2 ijerph-21-01122-t002:** Diarrhoeagenic bacteria’s identification in food and water samples collected from households of children with diarrhoea.

Diarrheagenic Pathogen	Steps for Strain Identification	Media	References
*Salmonella* spp./*Shigella* spp.	Pre-enrichment	BPW and mTSB broth	[10,11]
	Enrichment		
	Strain identification	XLD agar, HE agar, MAC agar, TSI agar, and MIO	
*Escherichia coli*	Enrichment	BPW	[12,13]
	Strins identification	MAC agar, TSI agar, and MIO	

BPW (Buffer Peptone Water, Liofilchem Diagnostici-2360611014, Roseto degli Abruzzi, Italy); mTSB (Modified Tryptic Soy Broth, Liofilchem Diagnostici-2360610352, Roseto degli Abruzzi, Italy); RVS (Rappaport-Vassiliadis with Soya, Liofilchem Diagnostici-610175, Roseto degli Abruzzi, Italy); MKTTn (Muller-Kauffman Tetrathionate-novobiocin Broth, Liofilchem Diagnostici-2360610239, Roseto degli Abruzzi, Italy); XLD (Xylose Lysine Deoxycholate agar, Neogen-NCM0021A, Lansing, MI, USA); HE (Hektoen enteric agar, HIMEDIA M377-500G, Kennett Square, PA, USA); MAC (MacConkey agar, Liofilchem Diagnostici-610028, Italy); TSI (Triple Sugar Iron agar, Liofilchem Diagnostici-360620055, Italy); MIO (Motility Indole Ornithine Medium, HIMEDIA–M378-500G, Square, PA, USA).

**Table 3 ijerph-21-01122-t003:** Sociodemographic characteristics of the caregivers and children under five years with diarrhoea (n = 186).

Variable	Category	Urban Area (n = 102)	Rural Area (n = 84)	*p*-Value for X^2^
		n (%)	n (%)	
Child gender	Female	42 (41.2)	43 (51.2)	0.22
	Male	60 (58.8)	41 (48.8)	
Caregiver gender	Female	97 (95.1)	78 (92.9)	0.74
	Male	5 (4.9)	6 (7.1)	
Child age (months)	≤6 months	11 (10.8)	22 (26.2)	0.02
	7 to 24 months	60 (58.8)	39 (46.4)	
	>24 months	31 (30.4)	22 (26.2)	
	Not disclosed	0	1 (1.2)	
Caregiver (age years)	18 to 25 years	50 (49.0)	42 (50)	0.94
	26 to 35 years	33 (32.4)	31 (36.9)	
	>35 years	10 (9.8)	9 (10.7)	
	Not disclosed	9 (8.8)	2 (2.4)	
Marital status	Single but lives with someone	75 (73.5)	67 (79.8)	0.26
	Single and living alone	10 (9.8)	3 (3.5)	
	Married	17 (16.7)	13 (15.5)	
	Widow	0	1 (1.2)	
Relation to the child	Mother	91 (89.2)	76 (90.5)	0.42
	Father	5 (4.9)	6 (7.1)	
	Grandmother/Parents	6 (5.9)	2 (2.4)	
Monthly income (USA Dollar)	0 to 70.00 $	6 (5.9)	7 (8.3)	0.85
	71.00–140.00$	9 (8.8)	8 (9.5)	
	>140.00$	2 (2.0)	2 (2.4)	
	Don’t know the income	34 (33.3)	24 (28.6)	
	Not disclosed	51 (50.0)	43 (51.2)	
Water source	Home pipe water	101 (99.0)	68 (81.0)	<0.05
	Public/community wells	0	12 (14.3)	
	Not disclosed	1 (0.5)	4 (4.8)	
Water treatment	Boiling	11 (10.8)	7 (8.3)	0.37
	Other treatment methods	1 (1.0)	1 (1.2)	
	Not disclosed	90 (88.2)	76 (90.5)	
Toilet system	Pit latrine with covering slab	28 (27.5)	12 (14.3)	<0.05
	Pit latrine without covering slab	3 (2.9)	13 (15.5)	
	Flush or pour toilet with septic tank, including squat toilet	58 (56.9)	27 (32.1)	
	Flush or pour toilet connected to sewer pipe	1 (1.0)	19 (22.6)	
	Other toilet system	0	3 (3.6)	
	Not disclosed	12 (11.8)	10 (11.9)	

**Table 4 ijerph-21-01122-t004:** Perception of the cause of diarrhoea by the caregivers of the children (n = 186).

Variable	Categories	Urban Area (102)	Rural Area (84)	*p*-Value for X^2^
		n (%)	n (%)	
Cause of diarrhoea in children under five years old	Vegetables	2 (2.0)	5 (6.0)	<0.05
	Cereals	2 (2.0)	3 (3.6)	
	Cookies/cake/popcorn	3 (2.9)	5 (6.0)	
	Fruit/Juice	3 (2.9)	9 (10.7)	
	Other food (peanut and fish)	1 (1.0)	1 (1.2)	
	Food not specified	4 (4.0)	2 (2.4)	
	Dentition	6 (5.9)	14 (16.7)	
	Weaning/Infant formula	5 (4.9)	15 (17.9)	
	Water	2 (2.9)	2 (2.4)	
	Other causes	2 (2.0)	3 (3.5)	
	Don’t know the reason	53 (52.0)	21 (25.0)	
	Not disclosed	19 (18.6)	4 (4.8)	

**Table 5 ijerph-21-01122-t005:** Food consumed by children under five a week before attending the health care centres.

Variables	Categories	Urban Area (n = 102)		Rural Area (n = 84)		*p*-Value for X^2^
		Consumed n (%)	Not Consumed n (%)	Consumed n (%)	Not Consumed n (%)	
Source of animal protein	Beef	23 (22.5)	78 (76.5)	64 (76.2)	19 (22.6)	<0.05
	Pork	11 (10.8)	90 (88.2)	45 (53.6)	38 (45.2)	<0.05
	Lamb	1 (1.0)	100 (98.0)	3 (3.6)	80 (95.2)	0.48
	Goat	3 (2.9)	98 (96.1)	27 (32.1)	56 (66.7)	<0.05
	Chicken	37 (36.3)	64 (62.7)	76 (90.5)	7 (8.3)	<0.05
	Egg	23 (22.5)	78 (76.5)	64 (76.2)	19 (22.6)	<0.05
	Not disclosed	1 (1.0)		1 (1.2)		N/A
	Fish	30 (29.4)	72 (70.6)	69 (82.1)	15 (17.9)	<0.05
Other sources of animal protein	Milk (pasteurised)	13 (12.7)	88 (86.3)	11 (13.1)	72 (85.7)	1.00
	Yoghurt	6 (5.9)	95 (93.1)	25 (29.8)	58 (6.9.0)	<0.05
	Cheese	3 (2.9)	98 (96.1)	0	83 (98.8)	0.32
	Not disclosed	1 (1.0)		1 (1.2)		N/A
Vegetables	Lettuce	6 (5.9)	40 (39.2)	9 (10.7)	13 (15.5)	<0.05
	Cassava leaves	12 (11.8)	34 (33.3)	10 (11.9)	12 (14.3)	0.19
	Pumpkin leaves	8 (7.8)	38 (37.2)	7 (8.3)	15 (17.9)	0.30
	Cowpea leaves	13 (12.7)	33 (32.4)	12 (14.3)	10 (11.9)	0.10
	Green cabbage	38 (37.3)	8 (7.8)	16 (19.0)	6 (7.1)	0.53
	Sweet potato leaves	6 (5.9)	40 (39.2)	3 (3.6)	19 (22.6)	1.00
	Cacana	0	46 (45.1)	7 (8.3)	15 (17.9)	<0.05
	Other vegetables	8 (7.8)	38 (37.3)	5 (6.0)	17 (20.2)	0.85
	Not disclosed	56 (54.9)		62 (73.8)		N/A
Fruits	Fruits	0	0	6 (7.1)	0	N/A
	Not disclosed	102 (100.0)		78 (92.9)		
Food consumed by children <2 years	Milk breastfeeding	25 (24.5)	77 (75.4)	19 (22.6)	65 (77.4)	0.90
	Infant formula	15 (14.7)	87 (85.3)	6 (7.1)	78 (92.9)	0.16
	Fermented cereals	22 (21.6)	80 (78.4)	28 (33.3)	56 (66.7)	0.10
	Ready-to-eat meals	7 (6.9)	95 (93.1)	8 (9.5)	76 (90.5)	0.69
	Fruit puree	12 (11.8)	90 (88.2)	11 (13.1)	73 (87.0)	0.96
	Biscuits/rusks/cookies	34 (33.3)	68 (66.7)	20 (23.8)	64 (76.2)	0.21
	Fruit/vegetable/juices	28 (27.5)	69 (67.6)	20 (23.8)	63 (75.0)	0.58
	Not disclosed	5 (4.9)		1 (1.2)		N/A
	Baby bottle	22 (21.6)	53 (52.0)	25 (29.8)	50 (59.5)	0.72
	Not disclosed	27 (26.5)		9 (10.7)		N/A
	Bottled water	22 (21.6)	79 (77.5)	23 (27.4)	61 (72.6)	0.48
	Not disclosed	1 (1.0)		0		N/A
	Boiled water	1 (1.0)	91 (89.2)	5 (6.0)	71 (84.5)	0.14
	Not disclosed	10 (9.8)		8 (9.5)		N/A
	Water not boiled	31 (30.4)	61 (59.8)	26 (31.0)	50 (59.5)	1.00
	Not disclosed	10 (9.8)		8 (9.5)		N/A

N/A: Not Applicable.

**Table 6 ijerph-21-01122-t006:** Pathogens distribution among food (n = 167) and water (n = 100) samples collected in under five children’s households.

Samples	Total number of isolates No (%)	Urban Area (n = 144)		Rural Area (n = 123)	
		DEC	*Salmonella* spp.	DEC	*Salmonella* spp.
		No (%)	No (%)	No (%)	No (%)
Cereal	9 (6.7)	4 (2.8)	1 (0.7)	3 (2.4)	1 (0.8)
Combined food	3 (2.4)	ND	ND	3 (2.4)	ND
Milk and Milk Products	1 (0.7)	1 (0.7)	ND	ND	ND
Vegetable (cooked)	ND	ND	ND	ND	ND
Other food (fish and fruit puree)	ND	ND	ND	ND	ND
Water	7 (5.4)	2 (1.4)	1 (0.7)	4 (3.3)	ND
Total	20 (15.2)	7 (4.9)	2 (1.4)	10 (8.1)	1 (0.8)

ND: Not detected; DEC: Diarrhoeagenic *E. coli.*

**Table 7 ijerph-21-01122-t007:** Diarrhoeagenic *E. coli* pathotypes prevalent in food and water samples collected in rural and urban households from children under five with diarrhoea.

Samples	Number of Isolates No (%)	Urban Area (n = 144)			Rural Area (n = 123)		
		ETEC No (%)	EPEC No (%)	EIEC No (%)	ETEC No (%)	EPEC No (%)	EIEC No (%)
Cereal	7 (5.2)	4 (2.8)	ND	ND	2 (1.6)	ND	1 (0.8)
Combined food	3 (2.4)	ND	ND	ND	3 (2.4)	ND	ND
Yoghurt	1 (0.7)	1 (0.7)	ND	ND	ND	ND	ND
Vegetable (cooked)	ND	ND	ND	ND	ND	ND	ND
Other food (fish and fruit puree)	ND	ND	ND	ND	ND	ND	ND
Water	6 (4.7)	1 (0.7)	1 (0.7)	ND	4 (3.3)	ND	ND
Total	17 (13.0)	6 (4.2)	1 (0.7)	0	9 (7.3)	0	1 (0.8)

EIEC—Enteroinvasive *Escherichia coli*; EPEC—Enteropathogenic *Escherichia coli*; ETEC—Enterotoxigenic *Escherichia coli;* ND—Not detected.

**Table 8 ijerph-21-01122-t008:** Binomial logistic regression of Diarrhoeagenic *E. coli* presence on food and water in urban and rural areas of Maputo.

Variable Group	Category	B (SE)	95% CI for Odds Ratio		
			Lower	Odds Ratio	Upper
Diarrheagenic *E. coli*					
	18 to 25 years	−3.15 (0.45)			
Caregiver age	26 to 35 years	−0.51 (0.67)	0.14	0.60	2.07
	>35 years	−0.42 (0.90)	0.83	0.66	3.28
Food consumed by children < 2 years	Fruit puree	1.50 (0.61) *	1.31	4.50	14.70
	Infant formula	1.33 (0.68) *	0.91	3.81	13.90
	Milk breastfeeding	−1.44 (0.08)	0.01	0.24	1.32
	Ready-to-eat meal	1.17 (0.68)	0.77	3.24	11.80
*Salmonella* spp.					
	Intercept	−4.60 (1.16) ***			
Caregiver age	26 to 35 years	−17.9 (>100)	N/A	<0.01	>100
Source of animal protein	Pork	−0.77 (1.43)	0.02	0.46	7.01
Food consumed by children < 2 years	Baby bottle	−0.03 (1.40)	0.13	0.97	14.5
	Bottled water	0.02 (1.32)	0.04	1.02	12.6
	Ready-to-eat meal	3.20 (1.31) *	2.03	24.6	>100
Kamakeni (Urban area)					
Caregiver age	18 to 25 years	−3.32 (0.69) ***			
	26 to 35 years	−1.04 (1.36)	0.01	0.3	3.71
	>35 years	0.09 (1.31)	0.04	1.09	11.5
Food consumed by children < 2 years	Bottled water	−18.67 (>100)	NA	<0.01	>100
	Infant formula	2.24 (1.08) *	0.04	9.38	83.4
	Ready-to-eat meal	2.03 (1.04) *	0.84	7.64	60.7
Marracuene (Rural area)					
	Male	−3.84 (1.41) **			
Caregiver gender	Female	−1.16 (1.33)	0.02	0.31	7.58
Food consumed by children < 2 years	Bottled water	3.04 (1.22) *	2.79	20.80	>100
	Baby bottle	−0.45 (1.21)	0.03	0.64	5.78
	Fruit puree	2.18 (0.87)*	1.65	0.80	56.20

Significance level: 0 ‘***’ 0.001 ‘**’ 0.01 ‘*’ 0.0; B: Estimated coefficient; E: Standard error; CI: Confidence interval; N/A: Not applicable

## Data Availability

The data presented in this study are available on request from the corresponding author as the data was stored electronically and transferred to a password-protected database to ensure privacy and confidentiality.

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
