# Peer review of "Diarrhoeagenic Escherichia coli and Salmonella spp. Contamination of Food and Water Consumed by Children with Diarrhoea in Maputo, Mozambique"

_ijerph, 2024, doi:10.3390/ijerph21091122_

Round 1

Reviewer 1 Report

Comments and Suggestions for Authors

The objective of this study was to identify Enterobacteriaceae contamination in food and water in urban and rural areas (page 1, line 15). In fact, this study examined only diarrheagenic E. coli and Salmonella. However, foodborne pathogens caused by Enterobacteriaceae include E. coli, Salmonella, Shigella, Vibrio, etc.

Please check the correction of bacterial culture media for Table 2 by following the FDA-BAM protocol (https://www.fda.gov/food/laboratory-methods-food/bam-chapter-4a-diarrheagenic-escherichia-coli). Modified Buffered Peptone Water with pyruvate (mBPWp) was used for the enrichment of E. coli (STEC pathotype).

Please include a description of the water samples in this study (bottled drinking water, tap water, well water, river water, etc.). The water samples should also be analyzed for the enumeration of fecal coliforms and E. coli. In general, the occurrence of fecal contamination is higher in environmental water than in sterile bottled water.

What is the procedure for water preparation from a 1-liter sample for the culture method? Did you use any filtration technique for concentration?

From the food (167) and water (100) samples analyzed by the culture method:

  • How many food and water samples were found to have E. coli colonies?
  • How many typical E. coli morphologies (i.e., pink color, circular, flat, entire margin, non-mucoid colonies on MacConkey agar) were picked up per sample for identification by biochemical test or MALDI-TOF?
  • Of the E. coli tested colonies, how many E. coli colonies were positive for a DEC pathotype? The author used a DEC PCR Kit (SSI) (page 4, line 124). This kit could identify DEC in each pathotype based on specific virulence genes. Please specify how many EPEC, ETEC, EIEC, and EHEC were found in each sample.
  • What are the most common/predominant pathotypes (EPEC, ETEC, EIEC, or EHEC) from food and water?
  • It would be better if serotyping and antimicrobial susceptibility testing are done with the DEC and Salmonella isolates. For example, Salmonella strains require serotyping, and some particularly prevalent serotypes such as S. typhi, S. typhimurium, and S. enteritidis need more precise identification.

Reviewer 2 Report

Comments and Suggestions for Authors

This manuscript determined the prevalence of diarrheagenic Escherichia coli and Salmonella in food and water collected in children’s households. Moreover, interviews were conducted to find the correlation with diarrheal pathogens. It provides some useful data and good research ideas. Some minor issues need to be modified.

1. When using the abbreviation "DEC" in the abstract, the full name should be given first.

2. The Latin name of bacteria needs to be italicized.

3. It is recommended to delete Figure 1 and Table 2 as they do not provide much useful information.

Comments on the Quality of English Language

good

Reviewer 3 Report

Comments and Suggestions for Authors

Since the authors used “diarrhoea” instead of “diarrhea,” diarrheagenic should also be spelt as “diarrhoeagenic.” See an example here: https://doi.org/10.1111/tmi.13735

Lines 14 and 36: age needs to be more specifically quantified (5 years or 5 months).

DEC was not spelt out in in the abstract.

Line 21: Does the statement “The prevalence of DEC and Salmonella spp. was 9.8% in food and 5.4% in water samples” mean that “Both DEC and Salmonella were present in 9.8% of food and in 5.4% of water samples," or “Either DEC or Salmonella was present in 9.8% of food and in 5.4% of water samples?" Or, did the authors mean "The prevalence of DEC and Salmonella spp. was 9.8% in food and 5.4% in water samples, respectively?"

Line 86: were the vegetables raw or cooked? Consuming the former would increase the risk of food-borne infections.

Line 86: It is also important to clarify if the milk given to children under 5 years underwent post-collection processing (e.g. pasteurization). Also, were all the milk of animal origin or were some human breastmilk?

Table 1: purity should be replaced with puree.

Line 100: spell out FDA BAM.

Line 119: state the MALDI-TOF MS score above which the identity given was deemed accurate/acceptable.

Tables 3 and 4: I believe the last column represents p-values and not chi-square values.

Also, are all the p-values in Table 4 less than 0.5, because there is only one value in the entire column?

Table 4: what is the difference between “food” and “fruits” since fruits are also a type of food? Probably the “food” category can be expanded to include meats, vegetables etc.

Please double check ALL the data in Table 5. A random check on the figures for fish consumed in the urban areas revealed a calculation error: when the 30 individuals who consumed fish were added with the 72 who did not, the total was already 102 individuals. Then what about the single individual who did not disclose his/her fish consumption? Adding him/her will result in a total of 103.

Tables 1 and 6 should contain the same sample categories. The “vegetable” and “other food” categories are missing from table 6. If no pathogen was detected for these two categories, ND should be stated.

Line 293: “purchasing power” seems inappropriately used in this context. A better term would be “lower socioeconomic status” or “lower income.”

Line 300: provide examples of pathogenic agents associated with poultry.

Comments on the Quality of English Language

Although the language used in the manuscript can be understood, it does contain grammatical errors. There is also an apparent poor attention to detail, as calculation mistakes were noted in one of the tables.

Round 2

Reviewer 1 Report

Comments and Suggestions for Authors
  1. Check abbreviations in Table 2: Ensure that the abbreviations "RVS" and "MKTT" are not included in Table 2.

  2. The discrepancy results were found in Table 6 and Table 7: Review the data carefully as there is a discrepancy in the number of DEC isolates from cereal in rural areas between the two tables.

  3. Lines 98 - 102: The revised sentence could be: "In lines 98 to 102, it is mentioned that 104 food and 37 water samples were found to contain Salmonella/Shigella isolates. MALDI-TOF was performed for confirmation using a single pure colony from each sample. My question is, from the 141 (104+37) Salmonella/Shigella isolates, why does Table 6 only show 3 isolates of Salmonella remaining? Does this mean that 138 isolates were Shigella?"

Reviewer 3 Report

Comments and Suggestions for Authors

Line 82: caw's or cow's milk?

Line 99: these 37 samples were water samples?

Table 8: abbreviations used [e.g. B(SE), CI] need to be spelt out in the footer.
